# Effects of Heat Stress on Estrus Expression and Pregnancy in Dairy Cows

**DOI:** 10.3390/ani15121688

**Published:** 2025-06-06

**Authors:** Szilvia Szalai, Ákos Bodnár, Hedvig Fébel, Mikolt Bakony, Viktor Jurkovich

**Affiliations:** 1Department of Animal Technology and Animal Welfare, Institute of Animal Sciences, Hungarian University of Agriculture and Life Sciences, 2100 Gödöllő, Hungary; 2Department of Obstetrics and Food Animal Medicine Clinic, University of Veterinary Medicine, 1078 Budapest, Hungary; febel.hedvig@univet.hu; 3Centre for Translational Medicine, Semmelweis University, 1085 Budapest, Hungary; bakony.mikolt@semmelweis.hu; 4Centre for Animal Welfare, University of Veterinary Medicine, 1078 Budapest, Hungary; jurkovich.viktor@univet.hu

**Keywords:** heat stress, estrus signs, behavior, hormone levels, pregnancy rate

## Abstract

This study examined how hot weather affects dairy cows’ ability to show signs they are ready to breed (called “estrus”) and how likely they are to become pregnant. Researchers worked with 58 healthy cows during summer and winter. The cows were given special hormone treatments to help them ovulate, and their behavior and hormone levels were closely monitored. In summer, when the weather was hot and stressful for the cows, they showed fewer signs of being ready to mate. For example, behaviors like standing still to be mounted or actively mounting other cows happened much less often, up to seven times less, compared to winter. As a result, it was harder for farmers to tell when cows were in estrus. Pregnancy rates also dropped sharply: only 10% of cows got pregnant in the summer, compared to 39% in the winter. One hormone treatment, G7G, helped maintain better pregnancy rates even in the heat, while another treatment worked well only in cooler weather. The study shows that hot weather can hamper fertility in dairy cows, but using the right hormone program and technology to track cow activity can help improve breeding success.

## 1. Introduction

Environmental factors, such as ambient temperature, humidity, and wind speed, can significantly affect the expression of estrus behavior of dairy cows [1]. The thermoneutral zone (TNZ) for high-producing dairy breeds, particularly Holstein Friesian, is generally considered to range from 5 °C to 25 °C [2] when animals generate basal levels of metabolic heat and have minimal physiological cost to reach maximum milk production and reproductive success [3]. Heat-stressed cows have reduced milk production, dry matter intake (DMI), and physical activity [4,5]. Reproductive capabilities of dairy cattle decline under heat stress (HS), as it disrupts multiple aspects of the reproductive process, including follicular development, steroidogenesis, oocyte quality, and embryo viability [6]. One of the primary mechanisms by which heat stress impairs fertility and inhibits reproductive performance in livestock is hormonal imbalance. Specifically, HS inhibits the pulsatile release of gonadotropin-releasing hormone (GnRH) via modulation of hypothalamic function, disrupting the hypothalamic–pituitary–gonadal axis. This disruption leads to altered secretion of key reproductive hormones, such as luteinizing hormone (LH) and follicle-stimulating hormone (FSH) [7]. Moreover, HS suppresses both the amplitude and frequency of LH pulses, compromising the maturation of dominant follicles and reducing estradiol synthesis [8]. A lower LH surge and tonic levels under HS conditions impair ovulation and the formation of functional corpus luteum (CL), leading to low progesterone levels [9]. The granulosa cells responsible for estradiol production are negatively affected by HS, which reduces estradiol production, resulting in silent expression of the estrus cycle in cattle [10]. A low fertility rate can be observed even in autumn due to the delay in the effect of HS on steroid production, leading to low-quality follicles and oocytes [11]. Folliculogenesis spans multiple estrous cycles; therefore, oocytes exposed to heat stress during summer may only ovulate several weeks later. Full recovery of oocyte developmental competence may require three to four estrous cycles [11]. HS can affect embryos at any stage; however, early-stage embryos are more susceptible. The preimplantation period is highly critical for the attachment of the developing embryo to the endometrium [12]. The intrauterine environment is also compromised in cows that suffer HS, including alterations such as diminished uterine blood flow and increased core body temperature [13].

In heat-stressed cows, the duration and intensity of estrus are changed; for example, activity and other manifestations of estrus are reduced [14], and the incidence of anestrus and silent ovulation is increased [15]. Cows can become acyclic under the influence of HS due to hyperthermia, oxidative stress, and physiological modifications in their body [16]. Automated sensor-based technologies that continuously monitor and record detailed information about the cow have been developed, which can improve the reproductive management of dairy farms [17]. López-Gatius et al. [18] detected a significantly lower increase in walking activity during summer. Expression of mounting activity is inhibited as long as the maximum environmental temperature rises above the thermoneutral zone. Beyond 30 °C, as observed by Gwazdauskas et al. [19], temperature negatively impacted the number of mounts. Reduction in LH secretion leads to suppressed synthesis of follicular steroids, thus reducing plasma estradiol concentrations, contributing to impaired estrus detection [4]. As HS affects reproductive performance, not only directly but indirectly, an alteration in energy balance can occur due to a reduction in dry matter intake. A negative energy balance leads to decreased plasma concentrations of insulin, glucose, and insulin-like growth factor 1 (IGF-1) and increased plasma levels of growth hormone and non-esterified fatty acids [20]. The metabolic hormone prolactin is temperature-sensitive, and its level increases in summer [21].

This study examined how environmental temperature affects estrus activity and pregnancy rates in lactating dairy cows during induced estrus. We hypothesized that as environmental temperatures increase, the expression of estrus behavior and pregnancy rate will decrease, and hormone levels will also change in heat stress periods.

## 2. Materials and Methods

### 2.1. Animals and Farm Conditions

The study was conducted on a large-scale dairy farm in Hungary (47°25′39.2″ N, 18°46′12.4″ E) with around 850 lactating Holstein Friesian dairy cows. Fifty-eight clinically healthy, multiparous, ovulation-synchronized cows were enrolled in this study under different weather conditions: summer (August 2023), under heat stress (HS; THImax = 80; *n* = 30), and winter (December 2023), no heat stress conditions (NHS; THImax = 48; *n* = 28). In both periods, cows were divided into G7G and OvSynch (OVS) groups according to the applied hormone program as follows: HSG7G *n* = 15 (days in milk (DIM): 68.1 ± 1.9; parity: 2.1 ± 1.2); HSOVS *n* = 15 (DIM: 160.7 ± 64.6; parity: 1.9 ± 0.6); NHSG7G *n* = 15 (DIM: 66.1 ± 1.6; parity: 2.7 ± 1.2); NHSOVS *n* = 13 (DIM: 157.3 ± 64.1; parity: 2.5 ± 1.3). The animals were housed in a free-stall barn with individual boxes bedded with straw and a grooved concrete floor equipped with fans above the feed line. Cows were milked twice daily at 4:00 and 16:00 in a Boumatic 2 × 2 × 12 herringbone milking system (BouMatic LLC, Madison, WI, USA). At the time of the experiment, the herd had a 305 d average milk yield of 9149 kg/cow. Cows received a TMR-formulated feed that met the nutritional requirements of lactating dairy cows, and it was delivered twice daily at approximately 5:00 and 14:00. Water and TMR were available for ad libitum intake.

### 2.2. Microclimate of the Barn

Weather data loggers (Voltcraft DL-181 THP; Conrad Electronic SE, Hirschau, Germany) were placed above the cows in the barn before the start of the ovulation synchronization program and recorded data until the pregnancy test during both study periods. The tool recorded the dry air temperature and relative humidity every 30 min from the beginning to the end of the study (a week before insemination until the pregnancy test, the 35th day after insemination). The following formula was used to calculate the temperature–humidity index (THI):THI = (0.8 × T) + (RH/100) × (T − 14.4) + 46.4
where T stands for dry air temperature (°C), and RH stands for relative humidity (%) [22]. The threshold value of heat stress was 68 [23,24], which is commonly used for high-producing dairy cows in continental regions.

### 2.3. Study Design

The cows were enrolled on one of the following oestrus synchronization protocols.

(1)Following the voluntary waiting period after calving, cows at 48 ± 3 DIM were enrolled in the G7G protocol, leading to their first timed artificial insemination (TAI). The G7G program was scheduled as follows (Figure 1): injection of PGF2α (cloprostenol; 0.250 mg, 2 mL, Syncoprost; Vetem S.p.A., Porto Empedocle, Italy) and a GnRH (gonadorelin 50 μg, 2 mL, Ovarelin; Ceva Sante Animale, Libourne, France) 2 d later, followed by a 7-day Ovsynch (OVS) protocol with double PGF, (GnRH, 7 d, PGF2α, 24 h, PGF2α, 32 h, GnRH, 16 h TAI) initiated 7 d later (Figure 1).

(2)After a negative pregnancy diagnosis, cows were enrolled in an OVS program (Figure 1) with double PGF (GnRH, 7 d, PGF2α, 24 h, PGF2α, 32 h, GnRH, 16 h TAI).

The G7G program TAI was on Thursday and the OVS program TAI was on Friday every week on the farm. On the day of insemination, estrus signs were scored (1–3) right before TAI. Two signs were chosen for evaluation: the amount of estrus mucus discharged and uterine erection. Score 1 = no/poor, score 2 = moderate, score 3 = a large amount of mucus/erect uterus. The quantity of mucus and uterine tonicity was assessed by rectal palpation. This well-established and widely used field method allows the evaluator to determine the presence of estrus and the firmness and contractility of the uterine wall. The uterus generally exhibits increased tone and rhythmic contractions during estrus due to elevated estrogen levels and heightened myometrial activity. The evaluation was conducted by the farm’s certified inseminator, in the presence of the first author, who is also highly experienced in the field. Sexed semen was used in primiparous cows undergoing their first post-calving artificial insemination (G7G protocol) in wintertime, while conventional semen was used in all other cases. Pregnancy diagnosis was made by chorioallantoic membrane slip technique through rectal palpation 35 days after TAI [25,26].

### 2.4. Samplings and Blood Analysis

After restraining the animals for TAI, blood samples were taken from the tail vessels into Li-heparin vacuum tubes (Vacuette, Greiner Bio-One, Kremsmünster, Austria). The samples were then cooled to 4 °C in a cooling bag and transported to the laboratory. After centrifugation at 4000 rpm for 10 min, the plasma samples were stored at −80 °C for further analysis. Estradiol, LH, prolactin, insulin, and IGF-1 were measured for hormone levels.

Estradiol (17β-estradiol) concentration of each sample was evaluated with ELISA (DE2693, Demeditec Diagnostics, Kiel, Germany; analytical sensitivity 10.6 pg/mL; intra-assay CV < 5%; inter-assay CVs were 14.9% and 6.9% for low and high controls, respectively).

Insulin was measured by a DE2935 ELISA test (Demeditec Diagnostics, Kiel, Germany). Analytical sensitivity was 2.99 uIU/mL. Intra-assay CV < 5%; inter-assay CVs were 5.04% and 0.16% for low and high controls, respectively.

IGF-1 was analyzed by a DEE020 ELISA test (Demeditec Diagnostics, Kiel, Germany). Analytical sensitivity was 0.813 ng/mL. Intra-assay CV < 5%; inter-assay CVs were 6.56%, 6.43%, and 5.53% for low, medium, and high controls, respectively.

Prolactin was measured with a bovine prolactin ELISA (EB0013, Wuhan Fine Biotech Co., Wuhan, China). Analytical sensitivity was 0.469 ng/mL. Intra-assay CV < 5%; inter-assay CVs were 5.36%, 5.12%, and 6.2% for low, medium, and high controls, respectively.

LH was measured by a DE1289 ELISA test (Demeditec Diagnostics, Kiel, Germany). Analytical sensitivity was 1.7 mIU/mL. Intra-assay CV < 5%; inter-assay CVs were 11.02%, 3.22%, and 4.45% for low, medium, and high controls, respectively.

### 2.5. Estrus Behavior Measurements

A day before their last injection of the synchronization protocol, cows were marked with a visible color number on their backs for observation through the Smart PSS (DH Vision Inc., Hanover, MD, USA) night vision camera system, which included 2 × 12 cameras covering the two groups. The system recorded videos and then downloaded them within 5 days for later analysis.

Besides the video surveillance, the activity of the animals was recorded by the Heat Seeker (BouMatic LLC, Madison, WI, USA) neck transponder, which transmitted the collected data to the Smart Dairy HerdMetrix (v. 1.20.1329; BouMatic LLC, Madison, WI, USA) system twice a day at milking times. The system indicates estrus based on the algorithm settings. However, to obtain a complete picture of the expression of estrus behavior in the last 12 h before TAI, 12 HS and 12 NHS cows from both synchronization programs (G7G, OVS) were analyzed from video. Mounting and standing behaviors were recorded for the marked cows in the last 12 h before TAI to analyze the estrus expression in different environmental conditions and then compare it with the result of hormone levels.

We were able to create three groups based on estrus signs. The animals that showed visible behavioral signs of estrus based on video or activity meter and had a large amount of estrus mucus on the day of TAI were classified as intensely estrus animals. The animals that did not show visible behavioral signs of estrus and did not have a large amount of mucus, but became pregnant, were classified as the silent estrus group. All other animals were considered non-estrus.

### 2.6. Statistical Analysis

We investigated the effect of the season besides the different hormone protocols. In the exploratory analysis, we mapped the patterns in the data using bar charts and average plots and examined the interactions between predictors.

For each type of hormone control, counts of pregnant and non-pregnant cows, and counts of animals in each of the estrus intensity categories, respectively, were compared between seasons using Fisher’s exact test.

Continuous variables (estradiol, luteinizing hormone, insulin, prolactin, and insulin-like growth factor-1) were investigated by linear models with the level of the given hormone as the response variable and type of hormone control, season, and their interaction term as fixed effects. Days in milk and parity were also included in the model as controlling variables. The amount of missing data for luteinizing hormone did not allow for control of these confounders.

In post hoc analyses, estimated means were compared between summer and winter for each hormone control group. The *p*-values were adjusted using the Tukey method for multiple comparisons. The level of statistical significance was determined at *p* < 0.05 for all tests.

Data visualization and hypothesis testing were performed in the R statistical environment (version 4.3.2) [27].

## 3. Results

### 3.1. Barn Microclimate

Table 1 shows the barn microclimate data during the study periods. Figure 2 displays that the daily maximum THI indices exceeded THI 68 throughout the summer study period, meaning that the cows were under heat stress.

### 3.2. Hormone Concentrations

Table 2 summarizes the hormone concentrations during the study periods. The average estradiol concentration, which is the hormone responsible for the expression of estrous behavior, showed equal values in the G7G program regardless of the season (*p* = 0.939; Table 2). In contrast, in the OVS program, it was significantly higher in winter compared to the summer period (Table 2). There were no significant differences in the concentrations of other hormones between the summer and winter periods (Table 2).

### 3.3. Estrus Behavior

Compared to the NHS period, heat-stress-related behavioral changes were observed in the HS group, especially estrus behavior. Most animals did not show estrus during the daytime in the heat-stressed period. Some cows walked intensively, followed others, and sniffed them, but typical standing and mounting behavior was not seen during the summer trial. It must be mentioned that some cows with spontaneous estrus were observed to mount each other, but this was only seen at night. Some cows in the trial started moving in the late afternoon and evening. Still, standing and mounting behavior could be observed in the morning when they would be milked and then wait for insemination. In contrast, most cows showed estrus behavior during winter, which started right after the last GnRH injection in the afternoon, a day before TAI. The number of standing and mounting behaviors also differed between the two seasons. In winter, five times more mounting (in total 125 in winter and 25 in summer; average ± SD/cow: 10.4 ± 12.4 in winter and 2.1 ± 4.0 in summer) and more than seven times more standing behavior (in total 124 in winter and 16 in summer; average ± SD/cow: 10.3 ± 11.6 in winter and 1.3 ± 2.5 in summer) could be observed through the cameras.

### 3.4. Estrus Intensity Changes in Heat Stress and Thermoneutral Period

In the G7G program, we observed more intensive estrus in animals in winter (Table 3). If we consider silent and intensive estrus animals as one group, the ratio of estrous animals is 53% in summer vs. 86% in winter. Almost half of the animals did not show estrus signs in the summer. Additionally, the number of silent estruses was higher; these cows did not show clear signs, so they were inseminated only because of the hormone program for TAI. Otherwise, they would have gone unnoticed by visual estrus observation. A significant difference was observed between the two seasons, even among animals that do not show signs of estrus. In the case of OVS, no significant difference in the number of cows in estrus was confirmed between the two seasons. However, the behavior was quite different from visual observation. Although the difference in estrus intensity was insignificant, the expression and time distribution of estrus behavior were significantly different. Silent estrus did not occur during winter, but we must consider the small number of animals.

### 3.5. Pregnancy Rate

In the winter, 39% of the animals that took part in the research became pregnant, while this ratio was only 10% in the summer, and all these animals were in the G7G hormone program (Table 3; Figure 3). In the case of the G7G program, there was no significant difference in pregnancy rate between winter and summer (Table 3; Figure 3). In contrast, significantly more cows from the OVS program became pregnant in winter than in summer (Table 3; Figure 3).

## 4. Discussion

Detection of estrus significantly affects the reproductive efficiency and profitability of dairy herds. Behavioral signs differ among individual cows in terms of duration and intensity of estrus, but factors such as environment and management also influence the expression of estrus. The focus on secondary symptoms of estrus may be more indicative than standing behavior, especially in the hot season when the behavior of the animals is altered. Possible signs of estrus include mounting behavior, increased physical activity, mucus discharge, swelling and reddening of the vulva, restlessness, bellowing, ruffled tailhead hair, and increased uterine contractility. Social interactions such as agonistic behavior, following, sniffing, licking, and chin-resting are also indicative [28]. Additionally, estrus may be accompanied by a sudden decrease in rumination, dry matter intake, water intake, and milk production [17]. Senger [29] established that follicles with a greater diameter on the day of estrus (≥12 mm) produce a higher amount of estradiol, which leads to increased cervicovaginal mucus production, resulting in a larger amount of visible estrus discharge. Wolfenson et al. [8] assumed high temperatures compromise endometrial function and alter its secretory activity. Schüller et al. [30] found that the amount of estrus discharge decreased continuously with increasing THI (≥63) on the day of estrus. Still, HS did not impact the intensity of uterine contractility.

The expression of estrus behavior depends on several factors, such as lameness, herd size, floor composition, climate, and the possibility of forming a sexually active group within a herd [31]. Orihuela [32] specified that environmental factors such as season, temperature, day length, and photoperiod have all been shown to influence estrus behavior in dairy cows. The leading cause of impaired heat detection is that HS reduces the steroidogenic capacity of theca and granulosa cells, essential for the ovarian follicles’ growth, maturation, and function, leading to diminished blood estradiol concentration [11]. In our study, the standing-mounting behavior was limited during the HS period, which agrees with the result of Sammad et al. [16] who observed that Holstein cows in estrus during summer exhibit an average of 4.5 mounts per estrus, whereas those in estrus during winter show 8.6 mounts per estrus and Sakatani et al. [33], who found that an extended period of high temperature shortens the duration and intensity of estrus signs. Furthermore, ovulations without estrous signs are more common during the warmer months [34]. In the case of some cows on the day of TAI, early morning estrus behavior was observed, but no mounting was seen during the daytime, only some intense walking and sniffing. In contrast, cows exhibited more expressed and much longer estrus behavior during the NHS period, and the pregnancy rate was almost four times higher. Morton et al. [35] estimated that a daily maximum THI of 72 or more, from day 35 before to day 6 after the day of breeding, decreases conception rates of lactating dairy cows by 30%. According to others, the conception rate of dairy cows is highly affected by HS, and this adverse effect begins at a THI ≥ 56 [36]. Depending on the study context, various THI thresholds for dairy cows have been reported in the literature, ranging from 56 to 72. We selected a threshold of 68 as a conservative indicator of moderate heat stress, appropriate for our regional conditions and dataset [23]. The significant reduction in pregnancy rate during summer reflects a strong heat stress effect; however, the seasonal difference is evident regardless of the specific threshold applied. Gilad et al. [37] found that HS causes a greater reduction in gonadotropin secretion in cows with low plasma estradiol than in those with high estradiol levels. This suggests that high estradiol may counteract the effects of HS, or the neuroendocrine system regulating gonadotropins is more sensitive to HS when estradiol is low. Reames et al. [38] suggest that the center in the hypothalamus responsible for behavioral estrus has a different activation threshold for estradiol than that for the LH surge. Therefore, at the end of a synchronization protocol, estradiol could be sufficient to induce a GnRH or LH surge but insufficient to induce behavioral estrus. Measurement of plasma concentrations of gonadotropins provides a good indication of the effects at these higher levels since the pulsatile secretion of LH is a reflection of the secretion of gonadotropin-releasing hormone (GnRH) from the hypothalamus [39]. Most studies report that HS decreases LH levels, so dominant follicles develop in a low LH environment, resulting in reduced estradiol secretion from the dominant follicle, poor estrus expression, and reduced fertility [40]. In our study, we did not measure plasma progesterone levels. Still, according to De Rensis and Scaramuzzi [4], it can be increased or decreased depending on whether HS is acute or chronic. Luteolysis could be caused by reduced estradiol synthesis and the failure of adequate estradiol secretion. This may lead to inadequate development of dominant follicles, resulting in smaller-sized follicles in the subsequent estrus [30]. There is no single mechanism by which HS reduces fertility, instead, this issue results from the accumulation of several factors. The plasma levels of insulin, IGF-I, and glucose decrease in summer compared to winter, probably due to lower dry matter intake and increased negative energy balance [41]. Insulin is required to develop follicles and benefits oocyte quality [42]. Prolactin is a metabolic hormone that is temperature sensitive, and its level increases in summer, which could inhibit follicular development [21]. These hormonal imbalances impair follicular development, hinder estrus detection, and reduce oocyte quality, compromising fertility.

Reproductive hormones are an alternative approach to improving summer fertility by stimulating ovarian function. The basic hormonal protocol is Ovsynch, where a luteolytic dose of PGF2α 6 follows GnRH 7 days later and a GnRH 48 h after luteolytic treatment to induce a fertile ovulation. Under HS conditions, this protocol showed no differences or decreased conception rates compared to cows inseminated at spontaneous estrus [35,40]. In our research, the farm used a second PGF treatment in Ovsynch-type protocols (G7G, OVS), which, according to Wiltbank et al. [43], can increase pregnancy success by about 10% in multiparous cows.

Fixed-time insemination has the distinct advantage of not requiring estrus detection. This approach increases the number of pregnant cows by increasing the number of inseminated cows. According to Yotov et al. [44], steroid hormone concentration on the day of TAI influences the pregnancy rates of dairy cows subjected to estrus synchronization and timed artificial insemination. The higher level of hormones may induce cyclicity and develop corpora lutea to maintain good fertility during the HS period [4]. In summer, synchronization programs do not increase the number of cows pregnant to fixed-time insemination. Still, they increase the number of cows pregnant within the optimal time postpartum and reduce the number of days open [8,45]. Cows that became pregnant had similar estrogen levels in the G7G group, also in NHS and HS conditions, while the OVS cows had lower estradiol levels in summer than in winter. The G7G program is longer and more complex, possibly resulting in more balanced hormone status and a higher estradiol level in the summer. This could also be seen in the pregnancy result because only cows in the G7G program had successful conceptions during the HS period. It should be noted that the results could be affected by the fact that cows in the OVS program had already been inseminated at least once or more. Another limitation, which could be a possible direction for future studies is that we could not measure a key hormone, progesterone, in our research. Also, measuring LH surge amplitude would add necessary information to future studies.

## 5. Conclusions

Heat stress negatively impacts estrus behavior in dairy cows, reducing standing and mounting behavior, and complicating visual estrus detection. While ovulation synchronization programs could help in specific cases, automated activity monitoring systems efficiently optimize reproduction. Monitoring additional health data, such as rumination, feed intake and estrus expression during the voluntary waiting period, may further enhance reproductive management. Maintaining conception rates is crucial for profitability, emphasizing the need for cooling systems to regulate body temperature and hormone balance. Future dairy operations should prioritize sustainability and environmental impact by increasing data-driven decisions and production efficiency while optimizing hormonal interventions to minimize dependence on treatments.

## Figures and Tables

**Figure 1 animals-15-01688-f001:**
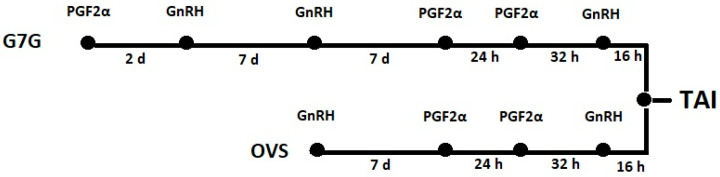
For Timed AI, the G7G and Ovsynch protocols were used with PGF2α and GnRH hormones. The G7G presynchronization started every 7 days, and non-pregnant cows found during pregnancy checks were synchronized in Ovsynch.

**Figure 2 animals-15-01688-f002:**
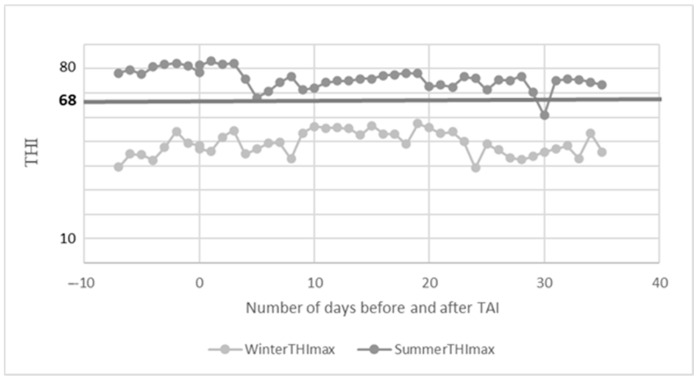
Temperature–humidity index (THI) throughout the study.

**Figure 3 animals-15-01688-f003:**
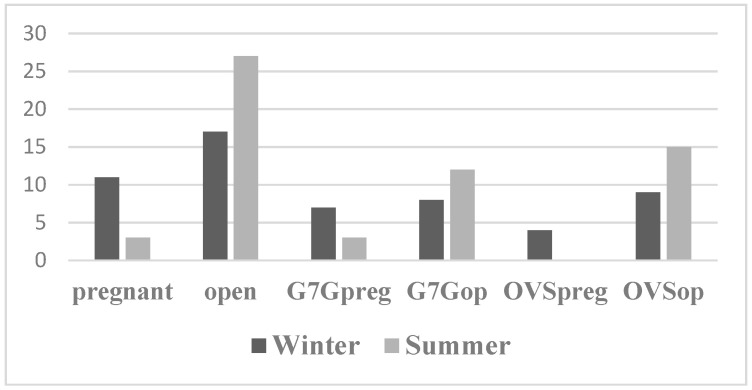
Pregnancy rate in different seasons and hormone programs. G7G: ovulation synchronization protocol, injection of PGF2α, and a GnRH, followed by a 7-day Ovsynch (OVS) protocol with double PGF. Preg: pregnant; OP: open (non-pregnant).

**Table 1 animals-15-01688-t001:** Barn microclimate data during the study periods.

	Summer	Winter
Ambient temperature (°C; average; min.; max.)	23; 14; 35	5; −4; 14
Relative humidity (%; average; min.; max.)	67; 32; 95	77; 39; 95
THI ^1^ (average; min.; max.)	70; 57; 83	44; 28; 57

^1^ THI: temperature–humidity index.

**Table 2 animals-15-01688-t002:** Hormone concentrations in the different groups during the study (with covariate ‘days in milk’ held at the overall mean value and averaged over the levels of ‘parity’).

	Summer Mean (95% CI ^1^)	Winter Mean (95%CI)	*p* Value
Estradiol (pg/mL)			
G7G ^2^	34.6 (24.8; 44.4)	38.2 (29.0;47.3)	0.525
OVS	18.9 (9.26; 28.6)	40.6 (31.3; 50.0)	<0.001
Luteinizing hormone (mIU/mL)			
G7G	0.54 (0.22; 0.86)	0.34 (0.08; 0.61)	0.322
OVS	0.25 (0.01; 0.62)	0.28 (0.01; 0.64)	0.928
Insulin (µIU/mL)			
G7G	20.8 (13.9; 27.8)	15.8 (8.74; 22.8)	0.2581
OVS	21.2 (14.0; 28.3)	21.2 (13.8; 28.7)	0.9914
Prolactin (ng/mL)			
G7G	32.4 (16.2; 48.5)	35.0 (19.7; 50.2)	0.7780
OVS	30.4 (13.6; 47.3)	37.1 (19.4; 54.8)	0.5023
Insulin-like growth factor 1 (ng/mL)			
G7G	121.0 (80.7; 161.0)	144.0 (106.8; 181)	0.3105
OVS	149.0 (110; 189)	183.0 (145.0; 221.0)	0.1800

^1^ CI: confidence interval; ^2^ G7G: ovulation synchronization protocol, injection of PGF2α and a GnRH, followed by a 7-day Ovsynch (OVS) protocol with double PGF.

**Table 3 animals-15-01688-t003:** The number of cows in estrus and the number of pregnant and non-pregnant cows during the study periods.

	Summer	Winter	*p*-Value ^1^
G7G ^2^			
intensive estrus	3	9	0.058
silent estrus	5	4	
no estrus	7	2	
OVS			
intensive estrus	9	10	0.154
silent estrus	4	0	
no estrus	2	3	
G7G			
pregnant	3	7	0.245
non-pregnant	12	8	
OVS			
pregnant	0	4	0.035
non-pregnant	15	9	

^1^ The effect of season on pregnancy; ^2^ G7G: ovulation synchronization protocol, injection of PGF2α and a GnRH, followed by a 7-day Ovsynch (OVS) protocol with double PGF.

## Data Availability

Dataset available on request from the authors.

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
