# Peer review of "Effects of Heat Stress on Estrus Expression and Pregnancy in Dairy Cows"

_animals, 2025, doi:10.3390/ani15121688_

Round 1

Reviewer 1 Report

Comments and Suggestions for Authors

The manuscript has some corrections that need to be made to improve the flow of the text.

Line 94 - add the meaning of the abbreviation "DIM". I understand that it is days in lactation, but it should be described in the text.

Line 134: describe how it was scored, did 1 or more people score it, was an average taken, or was some equipment used, and how was uterine tonicity assessed? Palpation or another method?

Tables: even if the acronyms are specified in the text, for better understanding of the tables, I suggest describing the meaning of the acronyms used in the footer.

Table 2 - the confidence interval can be described in the footer of the table so that the table has a better visual appearance. What is the unit of measurement of the hormones in this study?

Table 3 - "The effect of season on pregnancy", can be placed in the footer of the table, this improves the visual appearance of the table.

Author Response

Rev.: The manuscript has some corrections that need to be made to improve the flow of the text.
AU: Thank you for your valuable comments and suggestions. We tried to do our best to improve clarity.

Rev.: Line 94 - add the meaning of the abbreviation "DIM". I understand that it is days in lactation, but it should be described in the text.
AU: The abbreviation is explained.

Rev.: Line 134: describe how it was scored, did 1 or more people score it, was an average taken, or was some equipment used, and how was uterine tonicity assessed? Palpation or another method? 
AU: The text is supplemented with some explanation. 

Rev.: Tables: even if the acronyms are specified in the text, for better understanding of the tables, I suggest describing the meaning of the acronyms used in the footer.
AU: It is done accordingly.

Rev.: Table 2 – the confidence interval can be described in the footer of the table so that the table has a better visual appearance. What is the unit of measurement of the hormones in this study?
AU: CI is now explained in the footer. The units are supplemented in the table.

Rev.: Table 3 – „The effect of season on pregnancy”, can be placed in the footer of the table, this improves the visual appearance of the table.
AU: The table has been updated accordingly.

Reviewer 2 Report

Comments and Suggestions for Authors

Review comments/Report

In this manuscript titled “ Estrus behaviour, hormonal changes and pregnancy rate of dairy cows under heat stress,” the authors investigate how heat stress during summer affects estrus behavior, hormone levels, and pregnancy rates in dairy cows compared to winter conditions. The study evaluates the effectiveness of two hormonal synchronization protocols (G7G and OvSynch) and finds that heat stress significantly reduces visible estrus signs and conception rates, especially under the OvSynch protocol. The findings suggest that using the G7G protocol and automated monitoring systems may help mitigate the negative reproductive effects of heat stress. However, several sections require clarification, methodological rigor, and contextualization to strengthen the findings. Below are detailed comments organized by section. Therefore, Major revision is required before publication of this manuscript. The section wise comments are listed as;

Title & Abstract

Lines 2–3: The title accurately captures the study’s focus but could be more concise. Consider shortening to “Effects of Heat Stress on Estrus Expression and Pregnancy in Dairy Cows.”

Lines 26–40: The abstract is clear but mixes methods, results, and conclusions without transitions. Separate these logically or add clearer signal phrases (e.g., “We found that…”).

Introduction

Line 45: The thermoneutral range (5–25°C) lacks citation justification for dairy breeds used. Specify if this range is specific to Holstein-Friesians, given their sensitivity.

Lines 50–52: It affects the hypothalamus… is too generalized for this context. Specify the pathway, e.g., “Heat stress inhibits GnRH pulsatility through hypothalamic modulation.”

Lines 58–61: The timing of stress effects (“even in autumn”) is interesting but needs mechanistic insight. Briefly explain the lag due to disrupted folliculogenesis.

Material and Methods

Lines 91–96: Group descriptions are technically sound, but differences in parity and DIM across groups could confound outcomes. Include justification or statistical control methods used for these differences.

Lines 104–113: The THI formula is clearly presented but not explained. Add a line on why THI ≥68 is considered heat stress (already referenced in line 114).

Lines 117–122: The explanation of G7G protocol is dense and might overwhelm readers unfamiliar with AI protocols. Break into bullet points or a short table format for clarity.

Lines 132–135: Estrus scoring is subjective and sparsely defined. Define how mucus amount and uterine tone were assessed—by one observer or blinded multiple scorers?

Results

Lines 209–212: Statement “equal values in the G7G program regardless of season” lacks a strong statistical back-up. Provide exact p-values in text for transparency.

Line 230: “Five times more mounting…” is impactful but needs numeric values. State actual numbers (e.g., “5.2 mounts per cow in winter vs. 1.0 in summer”) for context.

Lines 245–250: “Only G7G cows conceived during summer” is a key point but not emphasized. Highlight this in both text and a clear summary table/figure.

Discussion

Line 260: “Secondary symptoms may be more indicative…” is valuable but should be justified. Cite observational studies or add from your own dataset.

Lines 284–286: Statement on THI ≥72 affecting conception rates is strong but needs cross-validation with current results. Discuss how the observed 10% summer pregnancy rate aligns with this threshold.

Line 299: A key hormone—progesterone—was not measured. Acknowledge this as a limitation and propose it for future studies.

Lines 332–335: Attribution of G7G success to “more balanced hormone status” is plausible but speculative. Suggest measuring LH surge amplitude and progesterone next time to confirm.

Conclusion (Lines 336–346)

Line 337: “Reducing intensity of external signs” is vague. Rephrase to “reducing standing and mounting behavior, complicating visual estrus detection.”

Line 344: The phrase “reduce hormonal and antibiotic treatments” lacks support from earlier data. Reframe to “optimize hormonal interventions to minimize dependence on treatments.”

General comments

  • Data Presentation: Strong data integration. However, figures should be referenced more often in the text.
  • Statistical Methods: Adequate but more emphasis on controlling for confounding (e.g., DIM and parity differences) would strengthen rigor.
  • Literature: Very thorough citations; includes current and classical studies.

Author Response

In this manuscript titled “ Estrus behaviour, hormonal changes and pregnancy rate of dairy cows under heat stress,” the authors investigate how heat stress during summer affects estrus behavior, hormone levels, and pregnancy rates in dairy cows compared to winter conditions. The study evaluates the effectiveness of two hormonal synchronization protocols (G7G and OvSynch) and finds that heat stress significantly reduces visible estrus signs and conception rates, especially under the OvSynch protocol. The findings suggest that using the G7G protocol and automated monitoring systems may help mitigate the negative reproductive effects of heat stress. However, several sections require clarification, methodological rigor, and contextualization to strengthen the findings. Below are detailed comments organized by section. Therefore, Major revision is required before publication of this manuscript. The section wise comments are listed as;

AU: Thank you for your valuable comments and suggestions. We accepted the recommendations and incorporated them into the manuscript to improve clarity and flow. All changes in the text are highlighted in yellow.

Title & Abstract
Rev.: Lines 2–3: The title accurately captures the study’s focus but could be more concise. Consider shortening to “Effects of Heat Stress on Estrus Expression and Pregnancy in Dairy Cows.”

AU: We changed the title according to the suggestion.

Rev.: Lines 26–40: The abstract is clear but mixes methods, results, and conclusions without transitions. Separate these logically or add clearer signal phrases (e.g., “We found that…”). 

AU: We changed the abstract accordingly.

Introduction

Rev.: Line 45: The thermoneutral range (5–25°C) lacks citation justification for dairy breeds used. Specify if this range is specific to Holstein-Friesians, given their sensitivity.

AU: We added a reference. 

Rev.: Lines 50–52: It affects the hypothalamus… is too generalized for this context. Specify the pathway, e.g., “Heat stress inhibits GnRH pulsatility through hypothalamic modulation.”

AU: We added lines to specify the pathway.

Rev.: Lines 58–61: The timing of stress effects (“even in autumn”) is interesting but needs mechanistic insight. Briefly explain the lag due to disrupted folliculogenesis. 

AU: We added a brief explanation.

Material and Methods
Rev.: Lines 91–96: Group descriptions are technically sound, but differences in parity and DIM across groups could confound outcomes. Include justification or statistical control methods used for these differences.

AU: The statistical analysis was repeated, controlling for DIM and parity.

Rev.: Lines 104–113: The THI formula is clearly presented but not explained. Add a line on why THI ≥68 is considered heat stress (already referenced in line 114). 

AU: Depending on the study context, various THI thresholds for dairy cows have been reported in the literature, ranging from 56 to 72. We selected a threshold of 68 as a conservative indicator of moderate heat stress, appropriate for our regional conditions and dataset [Reiczigel et al., 2009]. The significant reduction in pregnancy rate during summer reflects a strong heat stress effect; however, the seasonal difference is evident regardless of the specific threshold applied. We supplemented the text in the

Discussion.
Rev.: Lines 117–122: The explanation of G7G protocol is dense and might overwhelm readers unfamiliar with AI protocols. Break into bullet points or a short table format for clarity. 

AU: We believe that Figure 1 can explain the protocols

Rev.: Lines 132–135: Estrus scoring is subjective and sparsely defined. Define how mucus amount and uterine tone were assessed—by one observer or blinded multiple scorers? 

AU: We added some explanations in the text.

Results
Rev.: Lines 209–212: Statement “equal values in the G7G program regardless of season” lacks a strong statistical back-up. Provide exact p-values in text for transparency.

AU: It is provided.

Rev.: Line 230: “Five times more mounting…” is impactful but needs numeric values. State actual numbers (e.g., “5.2 mounts per cow in winter vs. 1.0 in summer”) for context. 

AU: We added numerical values. Also, we added a reference for this in the Discussion.

Rev.: Lines 245–250: “Only G7G cows conceived during summer” is a key point but not emphasized. Highlight this in both text and a clear summary table/figure

AU: Yes, pregnancy was 0 in summer in the OVS group, as shown in Table 3. However, we do not think that the two groups are fully comparable, since the initial conditions were different, as it is written in the Methods section. G7G was used for the first AI, while OVS was used after an unsuccessful G7G. 

Discussion

Rev.: Line 260: “Secondary symptoms may be more indicative…” is valuable but should be justified. Cite observational studies or add from your own dataset. 

AU: We added examples and references, too.

Rev.: Lines 284–286: Statement on THI ≥72 affecting conception rates is strong but needs cross-validation with current results. Discuss how the observed 10% summer pregnancy rate aligns with this threshold. 

AU: Depending on the study context, various THI thresholds for dairy cows have been reported in the literature, ranging from 56 to 72. We selected a threshold of 68 as a conservative indicator of moderate heat stress, appropriate for our regional conditions and dataset [23]. The significant reduction in pregnancy rate during summer reflects a strong heat stress effect; however, the seasonal difference is evident regardless of the specific threshold applied. We supplemented the text accordingly.

Rev.: Line 299: A key hormone—progesterone—was not measured. Acknowledge this as a limitation and propose it for future studies. 
AU: We added this to the limitations.

Rev.: Lines 332–335: Attribution of G7G success to “more balanced hormone status” is plausible but speculative. Suggest measuring LH surge amplitude and progesterone next time to confirm. 

AU: We supplemented the text accordingly.

Conclusion (Lines 336–346)
Rev.: Line 337: “Reducing intensity of external signs” is vague. Rephrase to “reducing standing and mounting behavior, complicating visual estrus detection.” 

AU: It is corrected.

Rev.: Line 344: The phrase “reduce hormonal and antibiotic treatments” lacks support from earlier data. Reframe to “optimize hormonal interventions to minimize dependence on treatments.” 

AU: It is corrected.

General comments
Rev.: Data Presentation: Strong data integration. However, figures should be referenced more often in the text.

AU: We referenced the tables and figures more in the text.

Rev.: Statistical Methods: Adequate but more emphasis on controlling for confounding (e.g., DIM and parity differences) would strengthen rigor.

AU: The statistical analysis for continuous variables was repeated with controlling for DIM and parity.

Round 2

Reviewer 1 Report

Comments and Suggestions for Authors I am grateful for the improvements made to the manuscript. The changes contributed to the clarity and organization of the text, making it well structured and suitable for publication.

Reviewer 2 Report

Comments and Suggestions for Authors

The authors have made substantial changes in the revised manuscript in view of points raised in the previous version.

However changes in conclusions section is needed by omitting the future suggestions. It should be based on major findings.